# *Conyza canadensis* from Jordan: Phytochemical Profiling, Antioxidant, and Antimicrobial Activity Evaluation

**DOI:** 10.3390/molecules29102403

**Published:** 2024-05-20

**Authors:** Lina M. Barhoumi, Ashok K. Shakya, O’la Al-Fawares, Hala I. Al-Jaber

**Affiliations:** 1Chemistry Department, Faculty of Science, Al-Balqa Applied University, Al-Salt 19117, Jordan; hala.aljaber@bau.edu.jo; 2Pharmacological and Diagnostic Research Center, Faculty of Pharmacy, Al-Ahliyya Amman University, Amman 19328, Jordan; 3Medical Laboratory Sciences Department, Faculty of Science, Al-Balqa Applied University, Al-Salt 19117, Jordan; ola.alfawares@bau.edu.jo

**Keywords:** *Conyza canadensis*, essential oil, GC/MS, phenolic compounds, flavonoids, LC-MS/MS, DPPH scavenging activity, antimicrobial activity

## Abstract

In this investigation, the chemical composition of the hydro-distilled essential oil (HD-EO), obtained from the fresh aerial parts (inflorescence heads (Inf), leaves (L), and stems (St)) of *Conyza canadensis* growing wild in Jordan was determined by GC/MS. Additionally, the methanolic extract obtained from the whole aerial parts of *C. canadensis* (CCM) was examined for its total phenolic content (TPC), total flavonoids content (TFC), DPPH radical scavenging activity, iron chelating activity and was then analyzed with LC-MS/MS for the presence of certain selected phenolic compounds and flavonoids. The GC/MS analysis of CCHD-EOs obtained from the different aerial parts revealed the presence of (2*E*, 8*Z*)-matricaria ester as the main component, amounting to 15.4% (Inf), 60.7% (L), and 31.6% (St) of the total content. Oxygenated monoterpenes were the main class of volatile compounds detected in the Inf-CCHD-EO. However, oils obtained from the leaves and stems were rich in polyacetylene derivatives. The evaluation of the CCM extract showed a richness in phenolic content (95.59 ± 0.40 mg GAE/g extract), flavonoids contents (467.0 ± 10.5 mg QE/ g extract), moderate DPPH radical scavenging power (IC_50_ of 23.75 ± 0.86 µg/mL) and low iron chelating activity (IC_50_ = 5396.07 ± 15.05 µg/mL). The LC-MS/MS profiling of the CCM extract allowed for the detection of twenty-five phenolic compounds and flavonoids. Results revealed that the CCM extract contained high concentration levels of rosmarinic acid (1441.1 mg/kg plant), in addition to caffeic acid phenethyl ester (231.8 mg/kg plant). An antimicrobial activity assessment of the CCM extract against a set of Gram-positive and Gram-negative bacteria, in addition to two other fungal species including *Candida* and *Cryptococcus*, showed significant antibacterial activity of the extract against *S. aureus* with MIC value of 3.125 µg/mL. The current study is the first phytochemical screening for the essential oil and methanolic extract composition of *C. canadensis* growing in Jordan, its antioxidant and antimicrobial activity.

## 1. Introduction

*Conyza* genus, a member of the *Asteraceae* (formerly known as Compositae) family, comprises more than 150 identified species of the flowering plants recognized as weeds in different types of crops and vineyards [1]. Despite their wide spread and distribution worldwide, the main origin of this genus is North and South America [2]. All species belonging to this genus are characterized by the production of a large number of seeds that can be dispersed by wind over long distances [3]. There are only five *Conyza* species reported in the flora of Jordan. These include *Conyza aegyptiaca* (L.) Aiiton, *Conyza albida* Sprigel, *Conyza bonariensis* (L.) Cronquist, *Conyza canadensis* (L.) Cronquist, and *Conyza stricta* [2].

*Conyza canadensis* L. Cronquist (Synonyms *Erigeron canadensis*), commonly known as horseweed or Canada fleabane, is an annual or perennial common weed native to North America [4]. The plant is recognized by its upstanding hairy long stem, growing up to 1.5 m tall. The plant branches above and overtopping its main axis with terminal cluster inflorescence, consisting of a large number of very small flower heads with a ring of white or pale purple ray florets and a center of yellow disc florets [5]. The flowering season extends from spring to autumn. The aerial parts of this weed are widely used in folk medicine to treat headache, dental pain, rheumatism, kidney problems, gastrointestinal disorders, and respiratory tract infections [6], in addition to wounds and skin burns treatments [7].

Literature surveys revealed that several *Conyza* species, including *C. canadensis*, were subjected to investigation of their volatile composition [4,5,8,9,10,11,12]. A previous phytochemical screening of *C. canadensis* secondary metabolites revealed the presence of polyacetylene derivatives [13], triterpenes [14,15,16], flavones, and phenolic compounds [17,18]. Furthermore, plants belonging to this genus were reported for their interesting bioactivity [4,6,8,19,20,21,22]. However, previous studies revealed that none of the *Conyza* plants from the flora of Jordan, including *C. canadensis*, were investigated before, neither for their chemical constituents nor for their bioactivity potentials. Accordingly, this investigation was designed to unveil the chemical composition and bioactivity potentials of *C. canadensis* from Jordan.

In this study, hydro-distillation (HD) was used to extract the essential oils (EOs) from the inflorescence heads (Inf), leaves (L), and stems (St) of natural populations of *C. canadensis* from Jordan. The different CCHD-EOs were then investigated for their chemical composition using the GC/MS technique. Moreover, the methanolic extract (CCM) obtained from the fresh aerial parts of *C. canadensis* was assayed for its total phenolic content (TPC), total flavonoids content (TFC), in vitro DPPH scavenging potential, and iron chelating activity. Additionally, the presence of a selected set of 14 phenolics compounds and 11 flavonoids was investigated by LC-MS/MS. The antimicrobial activity of CCM was also evaluated against a set of microbes including Gram-positive, Gram-negative bacteria, and fungi.

## 2. Results

### 2.1. GC/MS Analysis

HD-EOs obtained from the different fresh aerial parts of *C. canadensis* (Inf, L and St) were analyzed for their chemical constituents using GC/MS technique (Appendix A), results are listed in Table 1. The structures of the main constituents detected in the analyzed EOs are shown in Figure 1. Figure 2 reveals the main classes of volatile constituents detected in the analyzed oils.

### 2.2. Total Phenolics Content (TPC), Total Flavonoids Content (TFC), DPPH Radical Scavenging Activity, and Iron Chelating Activity

The CCM extract was assessed for its TPC, TFC, and antioxidant potential using the DPPH radical scavenging method and Fe^2+^ chelating activity (Table 2). Figure 3 displays the DPPH scavenging activity percent versus the concentration of CCM and two positive controls (ascorbic acid and α-tocopherol).

### 2.3. LC-MS/MS Analysis for Phenolic Compounds and Flavonoids

The CCM extract was also assayed using LC-MS/MS analysis, for the presence of selected 25 authentic compounds, comprising 14 phenolics and 11 flavonoids. The identity and concentration of the detected compounds (mg/kg plant) are shown in Table 3.

### 2.4. Antimicrobial Activity, Minimum Inhibitory Concentration (MIC), and Minimum Bactericidal Concentration (MBC) Determination

The agar-well diffusion method was used to assess the in vitro antifungal and antibacterial activity of CCM against three species of *Candida*, one species of *Cryptococcus*, four species of Gram-positive bacteria, and three species of Gram-negative bacteria. Around each well, the clear zone of inhibition (ZOI-mm) was measured.

The lowest concentration of CCM extract that inhibited bacterial growth was determined using the MIC assay, while the lowest concentration that killed 99.9% of the bacterial cells was calculated using the MBC assay. The ZOI, MIC, and MBC values are presented in Table 4.

## 3. Discussion

### 3.1. CCHD-EOs Data Analysis

The GC/MS analysis of the CCHD-EOs, obtained from the different aerial parts (Inf, L, and St), resulted in the identification of 109 compounds, of which, 64 were detected in the hydro-distilled oil obtained from the inflorescence heads, 56 in the leaves, and 54 in the stems. While the HDEOs of the different parts contained different classes of volatile compounds, all the EOs were characterized by their high polyacetylenes content, mainly the presence of (*Z*)-lachnophyllum ester and matricaria ester derivatives. In particular, (2*E*,8*Z*)-matricaria ester was the major component detected in the different oils in this study.

Oxygenated monoterpenes dominated Inf-CCHD-EO (38.79%) and was represented by limonene-1,2-diol (8.78%) and carvone (6.17%). The Inf-CCHD-EO contained also high concentration levels of polyacetylenes derivatives (27.00%) and monoterpene hydrocarbons (12.30%). These two classes contained mainly (2*E*,8*Z*)-matricaria ester (15.4%), limonene (11.9%), and (*Z*)-lachnophyllum ester (10.9%). Other classes of compound, including aromatic hydrocarbons, oxygenated aliphatic hydrocarbons, and fatty acids were detected in low concentration levels (0.85%, 0.23%, and 0.15%, respectively).

The essential oil obtained from *C. canadensis* leaves (L-CCHD-EO) was dominated by polyacetylenes derivatives that accounted for 68.90% of the total content. This class contained mainly (2*E*,8*Z*)-matricaria ester (60.7%) and its isomer, (2*Z*,8*E*)-matricaria ester (4.67%). Oxygenated sesquiterpenes amounted to 16.69% if the total composition and was represented by (*E*)-nerolidol (3.67%). Oxygenated monoterpenes (7.10%) contained mainly neryl acetone (1.86%). Both oxygenated aliphatic hydrocarbons (1.09%) and fatty acids (0.81%) in L-CCHD-EO were detected at slightly higher concentration levels as compared to their content in the oils obtained from the other two organs (Inf and St).

The analysis of the St-CCHD-EO revealed two major classes, including polyactelylene derivatives and oxygenated sesquiterpenes (39.80% and 34.53%, respectively). Again, (2*E*,8*Z*)-matricaria ester predominated the composition of St-CCHD-EO (31.6%). The oil contained other components like α-cadinol (5.70%), (*Z*)-lachnophyllum ester (4.01%) and (*E*)-nerolidol (3.71%). Interestingly, low concentration levels of the diterpenes (1.28%) were detected in St-CCHD-EO, which was completely absent in the CCHD-EOs of the other two parts.

Previous studies on the essential oil composition of several *Conyza* species revealed the detection of several volatile organic compounds with a wide spectrum of biological potential such as antibacterial, antioxidant, cytotoxic, anti-inflammatory, analgesic, antiviral, antiproliferative, and insecticidal activities [6,19]. Limonene was recognized as a major component detected in the EO of different *Conyza* species. This compound along, with other monoterpenes, detected in this species are known for their antioxidant, antimicrobial [23], and insecticidal activities [19]. Polyacetelenes derivatives, such as (Z)-lachnophyllum ester and matricaria ester isomers, were investigated for their antimicrobial [24] and antileishmanial [25] potentials. Studies on caryophyllene, caryophyllene oxide, and other sesquiterpenes revealed cytotoxic, anticancer, antioxidant, and antimicrobial properties [19]. Oxygenated monoterpenes were evaluated for antibacterial, antifungal, and antioxidant activity [26].

The essential oil of *C. canadensis* from Turkey, Pakistan, and Brazil, was dominated by limonene and polyacetylene derivatives, mainly (*Z*)-lachnophyllum ester and matricaria ester isomers [8,10,11,12]. In most reports, limonene dominated the EO obtained from whole aerial parts, while polyacetylene derivatives dominated the essential oil obtained from the roots [4,5,9]. Noticeably, the CCHD-EO of different aerial parts was rich in (2*E*,8*Z*)-matricaria ester isomer which was almost absent in EO of the plant from other regions. These results indicate clearly that the variation in the chemical composition of the essential oil was affected not only by the organ being investigated, but also by other environmental and experimental factors. These include soil properties, climate conditions, the time of harvesting, and extraction method. Table 5 summarizes the main variation in the HDEO composition of our study with the previous work.

### 3.2. TPC, TFC, DPPH Scavenging and Iron Chelating Activity for CCM Extract

As could be deduced from the data shown in Table 2, the CCM extract had relatively high DPPH scavenging power (23.75 ± 0.86 μg/mL) as compared to the tested positive controls (ascorbic acid: 1.79 ± 0.12 µg/mL; α-tocopherol: 5.00 ± 0.24 µg/mL). This result is mainly attributed to the high TPC and TFC in this extract (95.59 mg GAE/g extract, and 467 mg QE/g extract, respectively). In addition, the measured chelating effect of the extract on Fe^2+^ revealed a low chelating effect with IC_50_ 5396.07 ± 15.05 µg/mL compared to EDTA (20.15 ± 0.09 µg/mL).

Few studies reported the TPC and TFC for the methanolic plant extract of *C. canadensis*. The TPC and the TFC of *C. canadensis* from Moroccan origin [27] (2.54 µg /mg DM and 19.31 µg/mg DM, respectively) and Turkish origin [28] (71.34 ± 0.53 mg GAE/g extract; 18.91 ± 1.46 mg CA/g extract, respectively) were lower than those detected in our current investigation. Also, the observed DPPH radical scavenging power in our current study was higher than those observed in previous reports [20,27,28]. This could be mainly attributed to the high TPC and TFC detected in our study. Further confirmation was obtained upon LC-MS/MS analysis of this extract that revealed the detection of considerable concentration levels of rosmarinic acid, caffeic acid phenethyl ester, and apigenin-7-*O*-glucoside.

The antioxidant capacity of extracts obtained from *Conyza* genus was reported. Our research revealed interestingly moderate DPPH radical scavenging power as compared to other species from different geographical area (Table 6). In fact, the comparison of IC_50_ values of the different reports reveals the impact of environmental conditions on the chemical composition and antioxidant power.

### 3.3. LC-MS/MS Analysis for Phenolic Compounds and Flavonoids

LC-MS/MS profiling was performed to determine the presence of 25 compounds in the CCM extract. These included 14 phenolic compounds (vanillic acid, ascorbic acid, syringic acid, *p*-coumaric acid, ferulic acid, resveratrol, rosmarinic acid, salvianolic acid B, salvianolic acid A, chlorogenic acid, caffeic acid, gallic acid, carnosic acid, and caffeic acid phenethyl ester) and 11 flavonoids, including (catechin, hesperidin, apigenin-7-*O*-glucoside, hesperetin, rutin, quercetin, luteolin, apigenin, 3-*O*-methylquercetin, myricetin, and luteolin-7-*O*-glucoside). Of all these compounds, only four were not detected. These included caffeic acid, ascorbic acid, myricetin, and luteolin-7-*O*-glucoside.

The results revealed the detection of high concentration levels of rosmarinic acid (1441.1 mg/kg plant extract). Moreover, caffeic acid phenethyl ester was detected in moderate concentration levels (231.8 mg/kg extract). It is worth mentioning that this is the first report for the detection of 3-*O*-methylquercetin, hesperetin, resveratrol, salvianolic acid, hesperidin, and caffeic acid phenethyl ester in *C. canadensis*. Among the different flavonoids detected, it was noticed that apigenin-7-*O*-glucoside and 3-*O*-methylquercetin were the most abundant (46.13 and 36.64 mg/ kg plant, respectively). Trace amounts of each of chlorogenic acid, rutin, carnosic acid, gallic acid, and luteolin-7-*O*-glucoside were detected. The detection of high concentration levels of rosmarinic acid and other phenolics and flavonoids supports the observed DPPH radical scavenging power of the CCM extract obtained from *C. canadensis* from Jordan.

Rosmarinic acid was determined in a considerable concentration in our study. It was absent in the methanolic extract of Moroccan origin [22]. These findings further confirm the effect of environmental and climatic conditions on the biosynthetic pathways of plants, consequently leading to a wide spectrum of differences in secondary metabolite composition and bioactivity potentials.

### 3.4. Antimicrobial Assay

In this study, the CCM extract was assayed for its in vitro antifungal activity against three species of *Candida* (*Candida albicans*, *Candida krusei*, and *Candida glabrata*), one species of *Cryptococcus* (*Cryptococcus neoformans* (Sanfelice) vuillemin), and for its antibacterial activity against four species of Gram-positive bacteria (*Staphylococcus aureus*, *Staphylococcus hominis*, *Bacillus cereus*, and *Streptococcus pyogenes*), and three species of Gram-negative bacteria (*Salmonella typhi*, *Escherichia coli*, and *Pseudomonas aeruginosa*). The CCM extract was inactive at a concentration level of 100 ppm against all tested fungal species as compared to the positive control fluconazole. However, the extract revealed strong inhibitory effect against *S. aureus* with a minimum inhibitory concentration of 3.125 µg/mL (corresponding to MBC value of 6.25 µg/mL). The extract showed no interesting antibacterial activity against the other tested bacterial species at 100 ppm extract concentration. The characteristics of the microorganisms’ cell walls can be linked to the effectiveness of CCM extract as an antibacterial agent. For Gram-positive bacteria, teichoic acids make up over 60% of their cell wall [33]. In addition, Gram-positive bacteria only have one cell membrane, while Gram-negative bacteria have two: the outer and plasma membranes [34]. The outer membrane protects the bacterial cells from potentially hazardous substances by acting as a selective permeability barrier [35]. Furthermore, extracellular polymeric substances (EPS) that provide protection against harmful environmental conditions can be produced by Gram-negative bacteria, which could help and explain their resistance to a particular concentration of CCM extract [36].

The considerable antibacterial activity CCM extract against *S. aureus* may be attributed also to the high content of rosmarinic acid and caffeic acid phenethyl ester. The CCM extract obtained from the plant from Jordan showed higher inhibitory effects against the Gram-positive *S. aureus* (3.125 µg/mL) when compared to the alcoholic extract from Tunisian origin plant (MIC: 5 mg/mL) [20]. The extract obtained from the *C. canadensis* from Turkey showed even lower activity as compared to ours (ZOI: 7.0, 40.0 mm) [21]. It is worth noting that the extracts obtained from *C. canadensis* from Tunisian and Turkish origins both showed moderate antibacterial activity against *E. coli* [20,21]. Again, this variation is mainly attributed to the differences in the chemical composition resulting mainly from the effect of both environmental and climatic changes on the biosynthetic pathways in the plants.

*Staphylococcus aureus* is one of the main human pathogens that cause a wide variety of clinical illness. It is a leading cause of multiple human infection such as bacteremia, skin and soft tissue infections, pulmonary infections gastroenteritis, meningitis, and urinary tract infections. Treatment strategies are considered devastating due to the appearance of multi-drug resistant strains of species such as MRSA (Methicillin-Resistant *Staphylococcus aureus*) [37].

Several previous works have reported the antibacterial activity of different *Conyza* species against *S. aureus.* The comparison of the results obtained from the previous work with our current findings clearly indicated the significant antibacterial potentials of *C. canadensis* as compared to other *Conyza* species (Table 7). This could be attributed to the secondary metabolite composition and its effect on bioactivity. CCM could be a candidate as a plant-based drug for the treatment of infections caused by *S. arueses*. Table 7 summarizes the antibacterial effect (reported as MIC) observed for different *Conyza* species against *S. arueses*.

## 4. Materials and Methods

### 4.1. Plant Material

The aerial parts of the plant were collected from the Al Mansour neighborhood, Al-Jubeiha, Amman governorate, Jordan, during the autumn of 2023. The taxonomic identity of the plant was confirmed by Prof. Dr. Hala I. Al-Jaber, Department of Chemistry, Faculty of Science, Al-Balqa Applied University, Al-Salt, Jordan. A voucher specimen (No: Ast/Cc/2023) was deposited at the herbarium of the Faculty of Science (Natural Products Laboratory Herbarium), Al-Balqa Applied University, Al-Salt, Jordan.

### 4.2. Hydro-Distillation and Extraction of Essential Oils

Essential oils were extracted from fresh aerial parts of the of *C. canadensis* (inflorescence heads, leaves, and stems) according to the procedure described in the literature [40,41]. Briefly, a weighed sample of the specified fresh organ (Inf: 42.0 g, leaves: 86.21 g, stems: 107.0 g) was coarsely powdered and then subjected to hydro-distillation for 3 h in a Clevenger-type apparatus. The obtained essential oil (HDEO) from each organ was extracted (twice) with GC-grade *n*-hexane, dried using anhydrous Na_2_SO_4_, and then stored in an amber glass vials at 4 °C until analysis was performed. (% yield of the CCHD-EO: Inf: 0.38%, L: 1.71%, St: 0.023%).

### 4.3. GC-MS Analysis

GC/MS analysis was performed according to the procedure previously described in the literature [42,43]. The analysis was performed on Shimadzu QP2020 GC-MS equipped with GC-2010 Plus (Shimadzu Corporation, Kyoto, Japan) with split–splitless mode, utilizing a DB-5MS fused silica column (5% phenyl, 95% polydimethylsiloxane, 30 m × 0.25 mm, 25 µm film thickness). For the best component separation, a linear temperature program was used. Briefly, the oven temperature was set to 50 °C for 1 min, the temperature increasing from 50 °C to 280 °C, at a heating rate of 7 °C/min; then held at 280 °C for 10 min. The total run time was 44 min. The injector temperature was 260 °C with a split ratio of 20:1; an injection volume of 1 µL; a carrier gas: helium (flow rate 1.50 mL/min); and a flow control mode: pressure, 88.3 kPa. MS source temperature/detector temperature: 240 °C; interface temperature: 250 °C; ionization energy (EI): 70 eV; scan range 35–500 amu; scan speed 1666. The solvent cut was 3 min, while these data were acquired in 4.5 min. These data were collected using Windows based Lab-Solution GC-MS version 4.45SP1 Software. The mass spectra of isolated components were compared to those reported in ADAMS-2007 and NIST 2017 mass spectrometry libraries. To confirm the identified compound, a comparison performed between the reported values and relative retention indices (RI) with reference to *n*-alkanes (C_8_–C_30_) in addition to these data published in the literature [25,44,45].

### 4.4. Preparation of the Alcoholic Extract

The whole and fresh aerial parts of *C. canadensis* (20 g sample) were soaked in methanol (200 mL) at room temperature for 24 h as described in literature [42,46]. The procedure was repeated three times. The obtained extracts were combined and the solvent was then evaporated under reduced pressure at 55 °C. The obtained methanol extract (CCM yield: 7.63%) was then used for TPC, TFC, LC-MS/MS profiling, antioxidant activity evaluation, and bioactivity screening.

### 4.5. Total Phenolic Content (TPC) and Total Flavonoids Content (TFC)

The TPC and TFC were determined according the methods described in the literature [47] with slight modification. The Folin–Ciocalteu method was used to determine TPC. A 2.5 mL of Folin–Ciocalteu reagent (2N diluted ten folds) and 2 mL of Na_2_CO_3_ solution (75 g/L) were added to 0.5 mL of CCM (500 µg/mL). After incubating the solution for 1 h at room temperature, the absorbance of the resulting solution was measured at 765 nm. Methanol was used as a blank reference. Measurements were performed on Infinite M. Plex microplate reader (Tecan, Männedorf, Switzerland). The TPC of the CCM extract is reported as mg/g gallic acid equivalent. All measurements were performed in triplicates.

Briefly, the TFC was determined by diluting of 1.0 mL sample of extract (500 µg/mL) with 4.0 mL distilled water into a 10 mL volumetric flask and then 0.30 mL of the NaNO_2_ solution was added. After 5 min, 0.30 mL of the AlCl_3_ solution (10% *w*/*v*) was added to the mixture. The solution was incubated for 6 min, and then 2.0 mL of the 1.0 M NaOH solution was introduced and the final volume of the solution was adjusted to 10.0 mL with distilled water. After another 15 min, the absorbance of the resulting solution was measured at 510 nm using methanol as a blank. Measurements were performed on Infinite M. Plex microplate reader (Tecan, Männedorf, Switzerland). The TFC content in the plant extracts was determined and expressed in mg quercetin/g dry extract.

### 4.6. DPPH Free Radical Scavenging Activity

The free radical scavenging activity of CCM extract was determined by the 1,1-diphenyl-2-picryl-hydrazil (DPPH) according to the procedure described in the literature with minor modification [48]. Briefly, a volume of 1.0 mL from the prepared standard solutions at different concentrations (5–500 µg/mL) was added to 1.0 mL of the freshly prepared methanolic DPPH solution. After 30 min of incubation, the absorbance of the different solutions were measured at 517 nm. Ascorbic acid and α-tocopherol were used as positive control. A standard curve was prepared using different concentrations of the DPPH.

IC_50_ values of extract and standard were determined from the plot of scavenging activity against the compound’s concentrations, which were defined as the total antioxidant necessary to decrease the initial DPPH radical concentration by 50%. Experiments were carried out in triplicates. Measurements were performed on Infinite M. Plex microplate reader (Tecan, Männedorf, Switzerland).

### 4.7. Iron Chelating Activity

The chelating effect on ferrous ions by CCM was estimated by the method described by Sudan et al. with slight modifications [49]. Briefly, to 250 μL of the extract sample, 750 μL of methanol was added. Then, 50 μL of the 2 mM FeCl_2_ solution was added. The reaction was initiated by the addition of 100 μL of the 5 mM ferrozine into the mixture, which was then left at room temperature for 10 min. The absorbance of the mixture was determined at 562 nm.

### 4.8. LC-MS/MS Analysis of the CCM

The analysis of flavonoids and phenolic compounds was performed using a SciEx UPLC (Exion-UPLC, Framingham, MA, USA) equipped with the LC-ESI-MS/MS-4500-QTRAP system (AB Sciex Instrument, Framingham, MA, USA), utilizing Analyst 1.7 software for data analysis. As described in the literature [50], a chromatographic separation was conducted at 30 ± 1 °C using a Phenomenex column (Torrance, CA, USA, 3.0 × 50 mm, 5 µm). A gradient elution consisted of a mobile phase A (5 mM ammonium format in water: methanol (95:5; *v*/*v*)) and mobile phase B (methanol, 1 mM formic acid). The following ratio of mobile phase B was applied during gradient program (% B, min): 5–90% B (0.00–8.00 min), 90–90% (8.00–12.00 min), 90–5% (12.01–15.00 min). The solvent flow rate was 0.35 mL/min and the injection volume was 5 µL. MS/MS analysis was performed in positive and negative ion mode. Nitrogen gas was applied at a pressure of 60 psi as the nebulizing and drying gas. The mass spectra were obtained over an *m*/*z* range of 100–900 amu.

### 4.9. Strains, Media and Materials for Antimicrobial Activity

In vitro investigation of CCM for antimicrobial activity used four fungi strains (*Cryptococcus neoformans* (Sanfelice) vuillemin ATCC 32045, *Candida albicans* ATCC 10231, *Candida krusei* ATCC 6258, and *Candida glabrata* (Anderson) Meyer @ yarrow ATCC 20001). The investigation was conducted using Sabouraud dextrose broth (SDB) (Biolab, Budapest, Hungary). *Staphylococcus aureus* ATCC 25923, *Staphylococcus hominis* ATCC 27844, *Bacillus cereus* ATCC 14579, *Streptococcus pyogenes* ATCC 112696/19615, *Salmonella typhi* ATCC 13311, *Escherichia coli* ATCC 8739, and *Pseudomonas aeruginosa* ATCC 9027 were the studied strains of bacteria. Moller Hinton agar (MHA) from (Biolab, Hungary) was used.

### 4.10. Antifungal and Antibacterial Activity

The CLSI (Clinical and Laboratory Standards Institute) agar well diffusion method was used to evaluate the antibacterial activity of CCM [51]. On SDA, fungi were cultivated, and the mixture was incubated for 72 h at 25 °C. In the meanwhile, bacterial strains were grown on MHA and incubated for 24 h at 37 °C. Test strains of fungus (2 × 10^4^ CFU/mL-colony forming unit) and bacteria (~1 × 10^7^ CFU/mL) that matched a 0.5 McFarland were produced into a workable solution. In summary, 100 µliters of the suspension were applied individually to an SDA/MHA plate, and the suspension was then uniformly distributed throughout the agar’s surface. Next, using a 6 mm punching tool, wells were punched in each petri dish, and 100 µL of CC extract was added to each well. Fluconazole (10 µg/mL) and Ciprofloxacin (5 µg/mL) were used as a positive control for fungal and bacterial strains, respectively. The zone of inhibition around each well was measured in mm using a caliber. The assay was repeated and carried out in triplicate for each test isolate in a similar manner.

### 4.11. Assay for MIC and MBC

The minimal inhibitory concentration (MIC) of *C. canadensis* against *S. aureus* was determined by using the method of microdilution [52]. The bacterial strain was inoculated in Muller Hinton broth at 37 °C for 24 h. The turbidity of obtained cultures was adjusted to 0.5 Mcfarland. One hundred µL from diluted cultures were poured into a 96 well microtiter plate. Then, 100 µL of CCM extract stock solution was added to the first well followed by two-fold serial dilution to obtain different CCM concentrations (100, 50, 25, 12.5, 6.25, 3.13, 1.56, 0.78, 0.39, 0.195, 0.0975, and 0.04875 μg/mL). Then, the 96 well plate was incubated at 37 °C for 24 h. Visual examination of the incubated plate was performed by turbidity detection and changes observation. The optical density (OD) was measured at 600 nm using 96 well reader (Thermo-Scientific Multiskan SKY, Waltham, MA, USA). A control positive well was used, which had the tested culture and a negative control one that contained only sterile broth medium. The MIC defined as the least CCM concentration that inhibited bacterial growth after 24 h of incubation. A 50 μL from all wells that showed no visible growth or turbidity were cultivated on Tryptone Soya Agar (TSA), (Biolab, Hungary) and incubated at 37 °C for 24 h. The MBC (minimal bactericidal concentration) is known as the least CCM concentration that can prevent bacterial growth.

## 5. Conclusions

The increased widespread of *C. canadensis*, known also as horseweed, over the cultivated yards intensifies the interest to search for the weed biological activity and pharmaceutical application. We here report the phytochemical evaluation of the volatile composition and the alcoholic extract of *C. canadensis* from Jordan. Our current study revealed qualitative and quantitative variations in the composition of the HDEO of *C. canadensis* as compared to other previous studies. The detecting 2*E*,8*Z*-Matricaria ester isomer is reported at high concentration levels, which was almost absent in previous studies. Despite these changes, some similarities were observed. The richness of the CCHD-EOs in limonene and polyacetyelene derivatives encourages the future assessment of the oil for herbicidal and fungicidal activities. Moreover, the plant from Jordanian origin was found to be rich in rosmarinic acid as evidenced from its content in the CCM extract (1441.125 mg/kg plant). This is the first report for the detection and quantitation of 3-*O*-methylquercetin, hesperetin, resveratrol, salvianolic acid, hesperidin, and caffeic acid phenethyl ester in *C. canadensis*. Additionally, CCM also showed moderate antioxidant activity and significant antibacterial activity against *S*. *aureus* was detected.

## Figures and Tables

**Figure 1 molecules-29-02403-f001:**
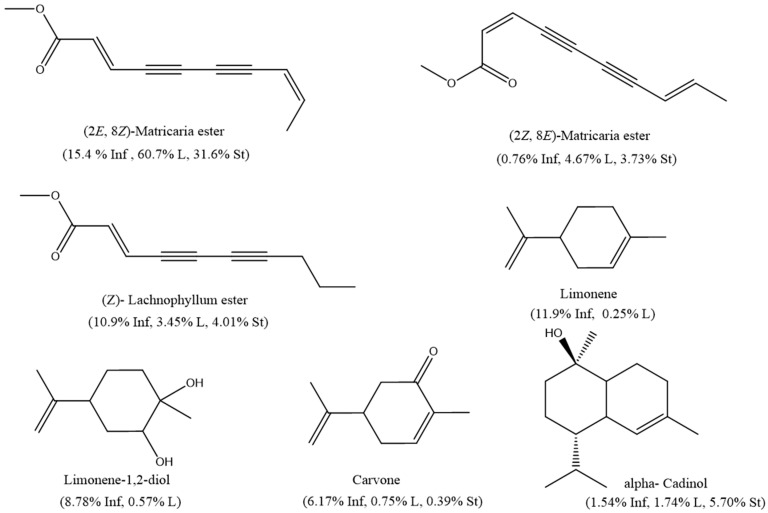
The structures of the main constituents detected in the CCHD-EO obtained from fresh aerial parts of *C. canadensis* from Jordan.

**Figure 2 molecules-29-02403-f002:**
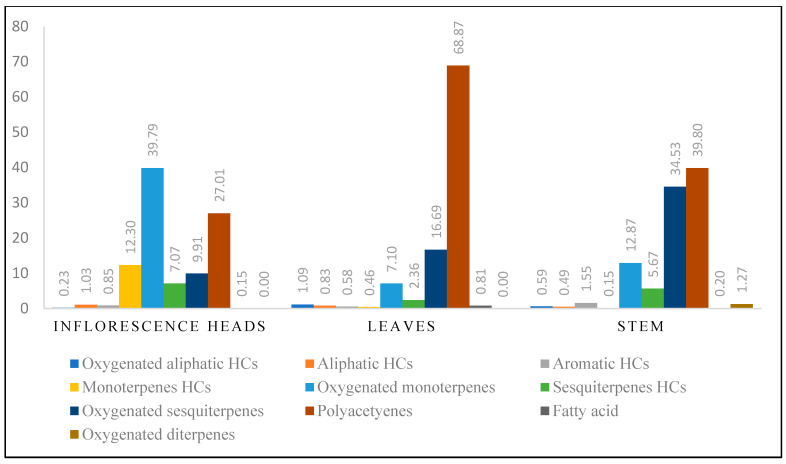
Different classes of the volatile compounds detected in CCHD-EOs obtained from the fresh aerial parts of *C. canadensis* from Jordan.

**Figure 3 molecules-29-02403-f003:**
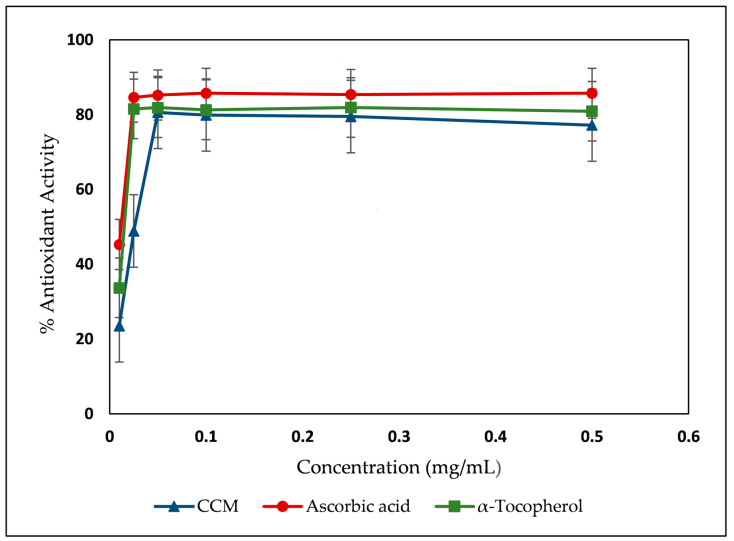
Antioxidant activity (%) by DPPH radical scavenging assay versus concentration (mg/mL).

**Table 1 molecules-29-02403-t001:** GC/MS analysis of the HDEO obtained from different parts of fresh *C. canadensis* grown in Jordan.

No.	Calc. RI	Lit. RI	Compound Name	Class	Molecular Formula	% Composition			Method of Identification
						Inf	L	St	
1	916	911	Isobutyl isobutyrate	OAHCs	C_8_H_16_O_2_	0.23	0.41	0.39	RI, MS
2	919	923	2-Methyl-4-heptanone	OAHCs	C_8_H_16_O	-	0.09	-	RI, MS
3	944	944	5-Mthyl-3-heptanone	OAHCs	C_8_H_16_O	-	0.12	-	RI, MS
4	947	968	*p*-Menthane	MHCs	C_10_H_20_	0.08	0.20	0.15	RI, MS, NIST
5	980	979	β-Pinene	MHCs	C_10_H_16_	0.18	-	-	RI, MS
6	1007	1008	*iso*-Sylvestrene	MHCs	C_10_H_16_	0.14	-	-	RI, MS
7	1026	1024	*p*-Cymene	ArHCs	C_10_H_14_	0.16	-	-	RI, NIST
8	1031	1029	Limonene	MHCs	C_10_H_16_	11.9	0.25	-	RI, MS
9	1125	1122	*trans*-*p*-Mentha-2,8-dien-1-ol	OMHCs	C_10_H_16_O	2.26	0.42	-	RI, MS
10	1134	1144	Limona ketone	OMHCs	C_9_H_14_O	0.46	-	-	RI, MS, NIST
11	1136	1142	*trans*-Limonene oxide	OMHCs	C_10_H_16_O	0.21	-	-	RI, NIST
12	1140	1137	*cis*-*p*-Mentha-2,8-dien-1-ol	OMHCs	C_10_H_16_O	2.84	0.42	-	RI, MS
13	1145	1139	*trans*-Pinocarveol	OMHCs	C_10_H_16_O	0.85	-	-	RI, MS
14	1167	1164	Pinocarvone	OMHCs	C_10_H_14_O	0.36	-	-	RI, MS
15	1172	1168	3-Thujanol	OMHCs	C_10_H_18_O	0.55	-	-	RI, MS
16	1182	1180	(*E*)-Isocitral	OMHCs	C_10_H_16_O	0.50	-	-	RI, MS
17	1191	1189	*trans*-*p*-Mentha-1(7),8-dien-2-ol	OMHCs	C_10_H_16_O	0.95	-	-	RI, MS
18	1195	1182	Isomenthol	OMHCs	C_10_H_20_O	-	-	0.34	RI, MS
19	1199	1188	α-Terpineol	OMHCs	C_10_H_18_O	-	0.43	0.51	RI, MS
20	1199	1192	*cis*-Dihydrocarvone	OMHCs	C_10_H_16_O	1.06	-	-	RI, MS
21	1202	1193	Dihydrocarveol	OMHCs	C_10_H_18_O	0.99	-	-	RI, MS
22	1203	1197	Verbanol	OMHCs	C_10_H_18_O	0.97	-	-	RI, MS
23	1209	1200	*trans*-Dihydrocarvone	OMHCs	C_10_H_16_O	0.34	-	-	RI, MS
24	1222	1216	*trans*-Carveol	OMHCs	C_10_H_16_O	5.04	0.61	-	RI, MS
25	1235	1229	*cis*-Carveol	OMHCs	C_10_H_16_O	1.58	0.16	-	RI, MS
26	1239	1230	*cis*-*p*-Mentha-1(7),8-dien-2-ol	OMHCs	C_10_H_16_O	0.19	-	-	RI, MS
27	1243	1237	Pulegone	OMHCs	C_10_H_16_O	-	-	3.43	RI, MS
28	1249	1243	Carvone	OMHCs	C_10_H_14_O	6.17	0.75	0.39	RI, MS
29	1274	1258	Carvenone	OMHCs	C_10_H_16_O	0.48	-	-	RI, MS
30	1281	1275	*p*-Mentha-1-en-7-al	OMHCs	C_10_H_16_O	0.50	-	-	RI, MS
31	1285	1289	Limonene-10-ol	OMHCs	C_10_H_16_O	0.24	-	-	RI, MS
32	1292	1294	Limonene diepoxide	OMHCs	C_10_H_16_O_2_	1.16	-	-	RI, MS, NIST
33	1304	1295	Perilla alcohol	OMHCs	C_10_H_16_O	0.38	-	-	RI, MS, NIST
34	1334	1342	*trans*-Carvyl acetate	OMHCs	C_12_H_18_O_2_	0.29	-	-	RI, MS
35	1345	1343	Piperitenone	OMHCs	C_10_H_14_O	-	-	1.63	RI, MS
36	1351	1343	Limonene-1,2-diol	OMHCs	C_12_H_18_O_2_	8.78	0.57	-	RI, MS, NIST
37	1358	1358	2,6-Dimethyl-2,7-octadiene-1,6-diol	OMHCs	C_10_H_18_O_2_	0.80	-	-	RI, MS, NIST
38	1365	1368	Piperitenone oxide	OMHCs	C_10_H_14_O_2_	-	-	2.84	RI, MS
39	1377	1370	*n*-Undecanol	OAHCs	C_11_H_24_O	-	-	0.19	RI, MS
40	1389	1381	β-Patchoulene	SHCs	C_15_H_24_	0.33	-	0.28	RI, MS
41	1393	1390	β-Elemene	SHCs	C_15_H_24_	0.18	-	-	RI, MS
42	1397	1392	(*Z*)-Jasmone	OMHCs	C_11_H_16_O	-	0.83	1.01	RI, MS
43	1410	1409	Citronellyl oxy-acetaldehyde	OMHCs	C_12_H_22_O_2_	0.66	-	-	RI, MS
44	1436	1434	*trans-*α-Bergamotene	SHCs	C_15_H_24_	4.27	1.23	2.82	RI, MS
45	1449	1436	Neryl acetone	OMHCs	C_13_H_22_O	1.16	1.86	1.73	RI, MS
46	1484	1480	ar-Curcumene	SHCs	C_15_H_22_	1.14	0.78	2.18	RI, MS
47	1489	1459	Sesquisabinene	SHCs	C_15_H_24_	1.01	0.35	0.39	RI, MS
48	1507	1497	Methyl *p*-tert butylphenyl acetate	ArHCs	C_13_H_18_O_2_	-	0.12	-	RI, MS
49	1510	1505	β-Bisabolene	SHCs	C_15_H_24_	0.13	-	-	RI, MS
50	1516	1511	(*Z*)-Lachnophyllum ester	PAcs	C_11_H_12_O_2_	10.9	3.45	4.01	RI, MS
51	1525	1527	(2Z,8*E*)-Matricaria ester	PAcs	C_11_H_10_O_2_	0.76	4.67	3.73	NIST
52	1530	1540	(2*E*,8*Z*)-Matricaria ester	PAcs	C_11_H_10_O_2_	15.4	60.7	31.6	NIST
53	1543	1548	(*E*)-Allyl cinnamate	ArHCs	C_12_H_12_O_2_	-	0.11	-	RI, MS
54	1549	1544	*cis*-Sesquisabinene hydrate	OSHCs	C_15_H_26_O	-	0.23	-	RI, MS
55	1560	1563	*epi*-Longipinanol	OSHCs	C_15_H_26_O	0.21	0.25	0.64	RI, MS
56	1563	1563	(*E*)-Nerolidol	OSHCs	C_15_H_26_O	1.33	3.67	3.71	RI, MS
57	1574	1564	Geranyl butanoate	OMHCs	C_14_H_24_O_2_	-	0.11	0.48	RI, MS
58	1577	1568	(*Z*)-3-Hexen-1-yl-benzoate	ArHCs	C_13_H_16_O_2_	-	0.10	-	NIST
59	1580	1597	(2*E*,8*E*)-Matricaria ester	PAcs	C_11_H_10_O_2_	-	-	0.43	NIST
60	1583	1583	ar-Turmerol	ArHCs	C_15_H_22_O	-	-	0.49	RI, MS
61	1585	1579	*trans*-Sesquisabinene hydrate	OSHCs	C_15_H_26_O	-	0.81	-	RI, MS
62	1586	1578	Spathulenol	OSHCs	C_15_H_24_O	-	-	3.08	RI, MS
63	1587	1569	Longipinanol	OSHCs	C_15_H_26_O	0.43	-	-	RI, MS
64	1589	1595	Cubeban-11-ol	OSHCs	C_15_H_26_O	0.31	-	-	RI, MS
65	1590	1592	Viridiflorol	OSHCs	C_15_H_26_O	-	0.09	-	RI, MS
66	1593	1583	Caryophyllene oxide	OSHCs	C_15_H_24_O	0.86	0.41	3.08	RI, MS
67	1596	1586	Presilphiperfolan-8-ol	OSHCs	C_15_H_26_O	-	-	0.19	RI, MS
68	1603	1596	Fokienol	OSHCs	C_15_H_24_O	-	0.53	0.62	RI, MS
69	1603	1602	Ledol	OSHCs	C_15_H_26_O	0.29	-	-	RI, MS
70	1612	1619	2,(7*Z*)-Bisaboladien-4-ol	OSHCs	C_15_H_26_O	0.33	0.83	0.85	RI, MS
71	1618	1607	Geranyl isovalerate	OSHCs	C_15_H_26_O	0.97	1.22	-	RI, MS
72	1625	1628	Isospathulenol	OSHCs	C_15_H_24_O	-	0.08	-	NIST
73	1635	1645	Cubenol	OSHCs	C_15_H_26_O	-	-	0.80	RI, MS
74	1637	1637	β-Acorenol	OSHCs	C_15_H_26_O	-	0.49	-	RI, MS
75	1643	1648	Khusilal	OSHCs	C_14_H_18_O	-	-	2.93	RI, MS
76	1650	1640	*tau*-Cadinol	OSHCs	C_15_H_26_O	-	1.73	2.91	RI, MS
77	1657	1653	Himachalol	OSHCs	C_15_H_26_O	-	0.64	1.72	RI, MS
78	1665	1654	α-Cadinol	OSHCs	C_15_H_26_O	1.54	1.74	5.70	RI, MS
79	1669	1658	α-Bisabolol oxide B	OSHCs	C_15_H_26_O_2_	0.21	0.38	0.82	RI, MS
80	1674	1679	(*Z)*-Methyl *epi*-jasmonate	OMHCs	C_13_H_20_O_3_	-	-	0.49	RI, MS
81	1678	1684	*epi*-α-Bisabolol	OSHCs	C_15_H_26_O	-	0.10	0.34	RI, MS
82	1681	1689	2,3-Dihydrofarnesol	OSHCs	C_15_H_28_O	0.31	0.48	1.38	RI, MS
83	1692	1685	α-Bisabolol	OSHCs	C_15_H_26_O	0.30	1.22	1.07	RI, MS
84	1696	1698	(2*Z*,6*Z*)-Farnesol	OSHCs	C_15_H_26_O	-	-	0.34	RI, MS
85	1695	1702	Sesquicineol-2-one	OSHCs	C_15_H_24_O_2_	0.88	-	-	RI, MS
86	1701	1700	Amorpha-4,9-diene-2-ol	OSHCs	C_15_H_24_O	-	0.19	-	RI, MS
87	1717	1713	Cedroxyde	OSHCs	C_15_H_24_O	-	1.32	0.34	RI, MS
88	1737	1729	γ-(*Z*)-Curcumen-12-ol	OSHCs	C_15_H_24_O	0.29	-	0.34	RI, MS
89	1745	1740	Oplopanone	OSHCs	C_15_H_26_O_2_	0.51	-	0.20	RI, MS
90	1752	1753	Xanthorrhizol	OSHCs	C_15_H_22_O	-	-	0.79	RI, MS
91	1780	1760	Benzyl Benzoate	ArHCs	C_14_H_12_O_2_	0.69	0.36	0.74	RI, MS
92	1808	1807	2-Ethylhexyl salicylate	ArHCs	C_15_H_22_O_3_	-	-	0.32	RI, MS
93	1829	1820	Isolongifolol acetate	OSHCs	C_17_H_28_O_2_	0.30	0.12	0.57	RI, MS
94	1836	1826	(*E*)-Nerolidyl isobutyrate	OSHCs	C_19_H_32_O_2_	0.34	0.17	0.51	RI, MS
95	1880	1887	Oplopanonyl acetate	OSHCs	C_17_H_28_O_3_	-	-	0.37	RI, MS
96	1895	1913	(5*Z*,9*E*)-Farnesyl acetate	OSHCs	C_18_H_30_O_4_	-	-	0.61	RI, MS
97	1892	1875	*n*-Hexadecanol	OAHCs	C_16_H_34_O	-	0.29	-	RI, MS
98	1897	1890	8-Isobutyryloxy isobornyl isobutanoate	OMHCs	C_18_H_30_O_4_	-	0.93	0	RI, MS
99	1920	1914	11,12-Dihydroxy valencene	OSHCs	C_15_H_26_O_2_	0.51	-	0.66	RI, MS
100	1921	1930	Musk ambrette	OAHCs	C_16_H_28_O_2_	-	0.18	-	RI, MS
101	1927	1921	Hexadecanoic acid, methyl ester	FA	C_17_H_34_O_2_	-	0.24	-	RI, MS
102	1966	1947	Isophytol	ODHCs	C_20_H_40_O	-	-	0.93	RI, MS
103	2096	2097	Linoleic acid, methyl ester	FA	C_19_H_34_O_2_	-	0.07	-	NIST
104	2102	2108	Linolenic acid, methyl ester	FA	C_19_H_32_O_2_	-	0.30	-	NIST
105	2113	2100	Heneicosane	AHCs	C_21_H_44_	-	0.74	0.49	RI, MS
106	2144	2133	Linoleic acid	FA	C_18_H_32_O_2_	0.15	0.21	0.20	RI, MS
107	2148	2149	Abienol	ODHS	C_20_H_34_O	-	-	0.35	RI, MS
108	2503	2500	*n*-Pentacosane	AHCs	C_25_H_52_	0.74	0.09	-	RI, MS
109	2704	2700	Heptacosane	AHCs	C_27_H_56_	0.29	-	-	RI, MS
		Oxygenated aliphatic hydrocarbons (OAHCs)				0.23	1.09	0.59	
		Aliphatic hydrocarbons (AHCs)				1.03	0.83	0.49	
		Aromatic hydrocarbons (ArHCs)				0.85	0.58	1.55	
		Monoterpenes hydrocarbons (MHCs)				12.30	0.46	0.15	
		Oxygenated monoterpenes (OMHCs)				39.79	7.10	12.87	
		Sesquiterpenes hydrocarbons (SHCs)				7.07	2.36	5.67	
		Oxygenated sesquiterpenes (OSHCs)				9.91	16.69	34.53	
		Polyacetylenes (PAcs)				27.00	68.90	39.80	
		Fatty acid (FA)				0.15	0.81	0.20	
		Oxygenated diterpenes (ODHCs)				-	-	1.28	
		Total Identified compounds				98.34	98.83	97.13	

**Table 2 molecules-29-02403-t002:** The total phenolic content (mg gallic acid/g extract), total flavonoids (mg quercetin/g extract), and DPPH radical scavenging activity in methanolic extract of *C. canadensis.* (Values expressed are means ± SD of three parallel measurements).

Sample	Yield	TPC	TFC	DPPH IC_50_ (μg/mL)	Iron Chelating Activity IC_50_ (µg/mL)
CCM	7.63%	95.59 ± 0.40	467.0 ± 10.5	23.75 ± 0.86	5396.08 ± 15.05
Ascorbic acid	-	-	-	1.79 ± 0.12	-
α-Tocopherol	-	-	-	5.00 ± 0.24	-
EDTA	-	-	-	-	20.15 ± 0.09

**Table 3 molecules-29-02403-t003:** LC-MS/MS data for the phenolic and flavonoid compounds detected in the CCM from Jordan and their concentrations.

No.	Compound Name	Molecular Formula	Structural Formula	*R*_t_ (min)	Concentration (mg Compound /kg Plant)
	Phenolics				
1.	Vanillic acid	C_8_H_8_O_4_	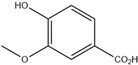	3.217	18.57
2.	Syringic acid	C_9_H_10_O_5_	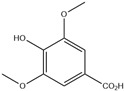	3.598	20.168
3.	*p*-Coumaric acid	C_9_H_8_O_3_	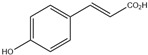	4.449	2.821
4.	Ferulic acid	C_10_H_10_O_4_	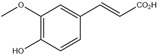	4.800	40.413
5.	Resveratrol	C_14_H_12_O_3_	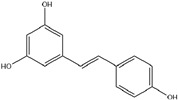	5.561	15.174
6.	Rosmarinic acid	C_18_H_16_O_8_	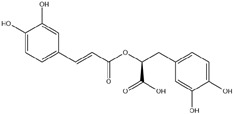	5.863	1441.125
7.	Salvianolic acid B	C_36_H_30_O_16_	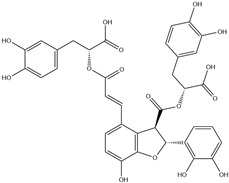	6.151	0.789
8.	Salvianolic acid A	C_26_H_22_O_10_	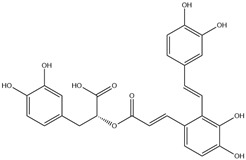	6.948	21.312
9.	Chlorogenic acid	C_16_H_18_O_9_	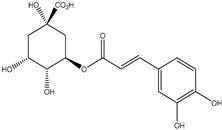	7.560	*t**
10.	Caffeic acid phenethyl ester	C_17_H_16_O_4_	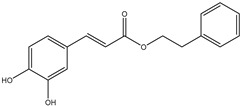	9.215	231.8
11.	Gallic acid	C_7_H_6_O_5_	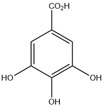	9.467	*t**
12.	Carnosic acid	C_20_H_28_O_4_	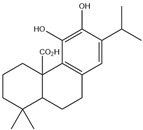	20.693	*t**
	Flavonoids				
1.	Catechin	C_15_H_14_O_6_	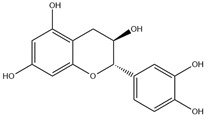	2.005	9.455
2.	Hesperidin	C_28_H_34_O_15_	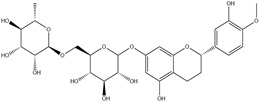	5.599	16.47
3.	Rutin	C_27_H_30_O_16_	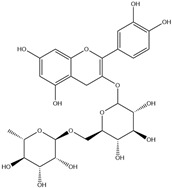	5.856	*t**
4.	Apigenin-7-*O*-glucoside	C_21_H_20_O_10_	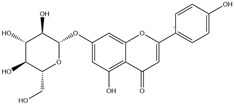	6.172	46.13
5.	Hesperetin	C_16_H_14_O_6_	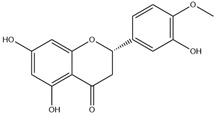	6.992	6.0612
6.	Quercetin	C_15_H_10_O_7_	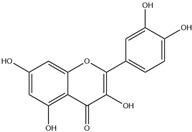	7.194	8.693
7.	Luteolin	C_15_H_10_O_6_	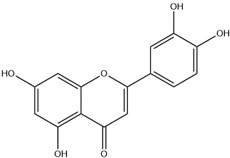	7.265	8.235
8.	Apigenin	C_15_H_10_O_5_	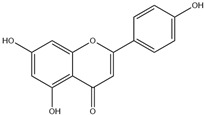	8.097	21.01
9.	3-*O*-Methylquercetin	C_16_H_12_O_7_	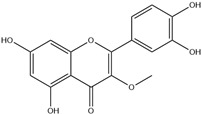	8.031	36.64

*t** = Trace amount.

**Table 4 molecules-29-02403-t004:** The ZOI-mm, MIC, and MBC of CCM against *S. aureus*.

Bacteria	Diameter of Inhibition Zone (mm)	MIC (µg/mL)	MBC (µg/mL)
*S. aureus*	40.0 ± 0.0	3.125	6.25

**Table 5 molecules-29-02403-t005:** The major components of essential oils from *C. canadensis* from different locations.

Compound Name	Pakistan [10]	Kashmir Valley (India) [8]	Pakistan [12]	China [11]	Turkey [5]		Hungary [4]		Brazil [9]		Present Study		
	AP	AP	AP	AP	AP	Rt	AP	Rt	L	Rt	Inf	L	St
β-Pinene	2.6	11.83	-	8.8	9.7	2.3	2.8	1.3	0.3	-	0.18	-	-
Limonene	41.3	23.78	28.4	41.5	28.1	0.9	79.2	1.0	38.0	-	11.9	0.25	-
*trans*-β-Ocimene	8.2	16.02	5.0	-	0.8	-	0.9	-	-	-	-	-	-
Carvone	-	-	-	3.8	0.5	-	-	-	1.2	-	6.17	0.75	0.39
*trans*-α-Bergamotene	2.7	2.07	3.6	-	0.8	-	2.9	trace	-	-	4.27	1.23	2.82
*cis*-Lachnophyllum ester	6.5	21.25	16.3	5.5	2.9	86.5	-	-	-	91.6	10.9	3.45	4.01
2*Z*,8*Z*-Matricaria ester	-	-	-	-	-	3.9	2.1	88.2	-	6.7	-	-	-
2*E*,8*Z*-Matricaria ester	-	-	-	-	-	0.5	0.3	1.9	-	-	15.4	60.7	31.6
Germacrene D	10.3	0.31	4.6	-	2.1	-	-	-	-	-	-	-	-
Spathulenol	-	0.18	-	-	16.3	2.0	0.3	-	10.7	-	-	-	3.08
Caryophyllene oxide	-	0.23	-	1.1	3.3	0.6	-	-	22.3	-	0.86	0.41	3.08
(*E*)-β-Farnesence	-	7.84	2.5	-	0.2	-	-	-	-	-	-	-	-
ar-Curcumene	-	2.99	-	-	0.3	-	-	-	-	-	1.14	0.78	2.18

AP: Aerial part, Rt: roots.

**Table 6 molecules-29-02403-t006:** The DPPH scavenging activity of *Conyza* spp. from different origins.

Conyza spp.	Location	DPPH, IC_50_ (µg/mL)	Reference
*C. canadensis*	Jordan	23.75 ± 0.86	Current study
*C. canadensis*	Morocco	88.19	[27]
*C. candensis*	Tunisia	120.0 ± 0.5	[20]
*C. aegyptiaca*	Cameroon	26.01 ± 1.09	[29]
*C bonariensis*	Pakistan	44.55	[30]
*C. dioscoridis*	Egypt	266.60	[31]
*E. alpinus*	India	38.75	[32]

**Table 7 molecules-29-02403-t007:** MIC values for different *Conyza* spp. extract against *S. arueses*.

*Conyza* Species.	Origin (Part)	MIC (µg/mL)	Reference
*C. canadensis*	Jordan, Arial part	3.125	Current study
*C. canadensis*	Tunisia, Aerial part	500	[20]
*C. dioscoridis* L.	Egypt, Aerial part	>800	[38]
*C. bonariensis*	Egypt, Aerial part	>800	[38]
*E. floribundus*	Cameroon, Leaves	512	[39]

## Data Availability

Data will be provided upon request.

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
