# Peer review of "Conyza canadensis from Jordan: Phytochemical Profiling, Antioxidant, and Antimicrobial Activity Evaluation"

_molecules, 2024, doi:10.3390/molecules29102403_

Round 1

Reviewer 1 Report

Comments and Suggestions for Authors

The authors in the current manuscript attempt to isolated an essential oil as well as a alcoholic fraction from Conyza canadensis. The authors have successfully performed an extensive (phyto)chemical characterisation, antioxidant and antimicrobial evaluation. Below I present some minor suggestions/ recommendation regarding this study.

Figure 1: I would encourage the authors to include underneath each structure the detected percentage of each structure within a sample.

Section 2.2: The authors should prepare a paragraph where they can actually cite Table 2 and not just mentioned Table 2. Additionally, the authors utilized DPPH assay in order to evaluate the direct radical quenching ability of CMM. I would encourage the authors to evaluate also the ability of CMM in the indirect radical scavenging, by means of investigating the chelation ability against redox active metals such as Cu2+ and Fe2+.
Figure 3. As the authors claimed, the experiment was performed in triplicates. Does this mean that the CMM isolation was performed 3times and each isolated fraction was analysed in triplicates? Or the same CMM fraction was assessed in triplicates? Please clarify. In any case a statistical analysis should be performed. Also, the error bars should be shown in the graph of Figure 3. I would also like to ask the authors why we cannot see the antioxidant evaluation of the isolated essential oil (please clarify and add relevant information).

Section 2.3
Since the authors utilised LC tandem mass spectroscopy for the quantification of polyphenolic compounds, chromatographic parameters including LOD, LOQ linearity accuracy, R2 of the calibration curve, recoveries and intra/inter precision should be presented in the supplementary material. In addition to this, the authors are encouraged to demonstrate which fragment has been used for quantification purposes and the extracted chromatograms of those fragment (again in a supplementary material), among with the calibration curves. Furthermore, in the case of utilising commercially available standards for the analysis information regarding the screened concentration, solvent of solubilisation, and product purchasing company/ product code should be included (in materials and methods). Furthermore, If I simply look at the 4.7 (material and methods), the authors stated that they used LC-MS/MS with a PDA detector. Therefore, information for the wavelength that each compound has been identified should also be stated.  

Table 4: Despite the fact that the authors claimed that they screen the antibacterial and antifungal anticity of overall 4 species, table 4 show data only for 1 (S. aureus). Please clarify or make any relevant corrections. I would also like to ask the authors why we cannot see the antimicrobial/antifungal evaluation of the isolated essential oil (please clarify and add relevant information).

Line 318: please substitute organs with parts.

Line 387 should be Switzerland

Line 86 C canadensis should be italics

Line 114 remove extra bracket ((mg/ kg plant)

Please keep consistent between mL and ml.

Table 1: please substitute organs with aerial parts and re-write the species using italics

Author Response

Dear editor in chief

First, on behalf of all coauthors, I would like to thank you very much for your great efforts in this highly esteemed Journal. We wish also to thank all the reviewers for their valuable comments that helped in improving our manuscript. All comments were considered.

Please note the following:

Response to Reviewer 1

 Comment: Figure 1: I would encourage the authors to include underneath each structure the detected percentage of each structure within a sample.

Answer: The comment was considered and corrected.

Comment: Section 2.2: The authors should prepare a paragraph where they can actually cite Table 2 and not just mentioned Table 2. Additionally, the authors utilized DPPH assay in order to evaluate the direct radical quenching ability of CMM. I would encourage the authors to evaluate also the ability of CMM in the indirect radical scavenging, by means of investigating the chelation ability against redox active metals such as Cu2+ and Fe2+. 

Answer: Done. The paragraph was corrected according to the reviewer comment. The metal chelating ability of the tested extract was evaluated and the results were included in the manuscript.

                In addition, the nitric oxide radical scavenging activity was evaluated but we did not receive any results hence it was not included in the manuscript.

Comment: Figure 3. As the authors claimed, the experiment was performed in triplicates. Does this mean that the CMM isolation was performed 3times and each isolated fraction was analyzed in triplicates? Or the same CMM fraction was assessed in triplicates? Please clarify. In any case a statistical analysis should be performed. Also, the error bars should be shown in the graph of Figure 3. I would also like to ask the authors why we cannot see the antioxidant evaluation of the isolated essential oil (please clarify and add relevant information).

Answer: We would like to point out that the same extract for evaluating the antioxidant activity. The antioxidant activity assay was performed in triplicates using the same extract. The error bars were added to Figure 3.

  • The yield of HDEO was low and insufficient to perform the antioxidant activity.

Comment: Section 2.3 Since the authors utilized LC tandem mass spectroscopy for the quantification of polyphenolic compounds, chromatographic parameters including LOD, LOQ linearity accuracy, R2 of the calibration curve, recoveries and intra/inter precision should be presented in the supplementary material. In addition to this, the authors are encouraged to demonstrate which fragment has been used for quantification purposes and the extracted chromatograms of those fragment (again in a supplementary material), among with the calibration curves. Furthermore, in the case of utilizing commercially available standards for the analysis information regarding the screened concentration, solvent of solubilization, and product purchasing company/ product code should be included (in materials and methods). Furthermore, If I simply look at the 4.7 (material and methods), the authors stated that they used LC-MS/MS with a PDA detector. Therefore, information for the wavelength that each compound has been identified should also be stated.

Answer: Done. The comment was considered. The LOD, LOQ, range, fragment ions and supplier are mentioned in the supplementary Table S1. The selected authors are preparing a separate analytical paper; hence all the details are not included in the supplementary data. The material and methods are updated.

  • PDA was configured with LC-MS/MS but it was not used in analysis.

Table 4: Despite the fact that the authors claimed that they screen the antibacterial and antifungal activity of overall 4 species, table 4 show data only for 1 (S. aureus). Please clarify or make any relevant corrections. I would also like to ask the authors why we cannot see the antimicrobial/antifungal evaluation of the isolated essential oil (please clarify and add relevant information).

Answer: The extract showed no interesting antifungal activity against all tested species. Also, it was inactive against the other tested bacterial species (please check the figure below for the observed inhibitor zones related to our study, this figure was not included in the manuscript but was added to supplementary material).

  • Unfortunately, the limited amount of essential oil collected by hydro-distillation did not allow us to further investigation for its antibacterial / antifungal activities against the indicated species.

- All the mentioned typographical mistakes were corrected, and highlighted in the revised manuscript.

Reviewer 2 Report

Comments and Suggestions for Authors

I consider that the paper entitled “Conyza canadensis from Jordan: Phytochemical Profiling, Antioxidant and Antimicrobial Activity Evaluation” could be published in the journal Molecules, after making a notable improvement in the discussion of results in the following aspects:

1.     The greatest contribution of this paper is the identification of various chemical compounds, some with bioactive properties, which were extracted from a plant (Conyza canadensis) recognized as a weed of different crops and vineyards. In this context, I believe that there is a need for a broad discussion on the bioactive properties (antioxidant and/or antimicrobial) that the major compounds could provide, including matricaria ester derivatives, oxygenated monoterpenes and polyacetylene derivatives, because the antioxidant activity of polyphenols and flavonoids, on which the authors focused, has already been documented even for other species of this plant family.

2.     Based on the fact that the antioxidant activity of the CCM extracts was only evaluated by the potential for inhibition of the DPPH radical, it would be advisable to include a table to compare the IC50 values obtained in this study with those reported for other species of this plant and even in other plant tissues, to highlight their antioxidant potential.

3.     The activity of the CMM extracts was also evaluated against a set of microbes, including gram-positive bacteria (Staphylococcus aureus, Staphylococcus hominis, Bacillus cereus, Streptococcus pyogenes), gram-negative bacteria (Salmonella typhi, Escherichia coli, Pseudomonas aeruginosa) and fungi (Candida albicans, Candida krusei, Candida glabrata, Cryptococcus neoformans). Because only a strong inhibitory effect was found against Staphylococcus aureus, I suggest including a table comparing the minimum inhibitory concentration values of these extracts against other species of this plant and even in other plant tissues, to highlight their antimicrobial potential. Likewise, highlight the type of diseases caused by Staphylococcus aureus, which could be combated by the use of CMM extracts.

4.     Additionally, it would be advisable to improve the wording of the conclusions. The results shown do not support the following conclusion: Besides, a considerable antioxidant activity and significant antibacterial activity of the methanolic extract against S. aureus were detected. The antioxidant activity evaluated only against the DPPH radical is not sufficient for such a conclusion.

Author Response

Dear editor in chief

First, on behalf of all coauthors, I would like to thank you very much for your great efforts in this highly esteemed Journal. We wish also to thank all the reviewers for their valuable comments that helped in improving our manuscript. All comments were considered.

Please note the following:

Response to Reviewer 2:

I consider that the paper entitled “Conyza canadensis from Jordan: Phytochemical Profiling, Antioxidant and Antimicrobial Activity Evaluation” could be published in the journal Molecules, after making a notable improvement in the discussion of results in the following aspects:

Comment 1: The greatest contribution of this paper is the identification of various chemical compounds, some with bioactive properties, which were extracted from a plant (Conyza canadensis) recognized as a weed of different crops and vineyards. In this context, I believe that there is a need for a broad discussion on the bioactive properties (antioxidant and/or antimicrobial) that the major compounds could provide, including matricaria ester derivatives, oxygenated monoterpenes and polyacetylene derivatives, because the antioxidant activity of polyphenols and flavonoids, on which the authors focused, has already been documented even for other species of this plant family. Answer: Done.

Comment 2: Based on the fact that the antioxidant activity of the CCM extracts was only evaluated by the potential for inhibition of the DPPH radical, it would be advisable to include a table to compare the IC50 values obtained in this study with those reported for other species of this plant and even in other plant tissues, to highlight their antioxidant potential.

Answer: Done. The required table was added (table 6).

Comment 3: The activity of the CMM extracts was also evaluated against a set of microbes, including gram-positive bacteria (Staphylococcus aureus, Staphylococcus hominis, Bacillus cereus, Streptococcus pyogenes), gram-negative bacteria (Salmonella typhi, Escherichia coli, Pseudomonas aeruginosa) and fungi (Candida albicans, Candida krusei, Candida glabrata, Cryptococcus neoformans). Because only a strong inhibitory effect was found against Staphylococcus aureus, I suggest including a table comparing the minimum inhibitory concentration values of these extracts against other species of this plant and even in other plant tissues, to highlight their antimicrobial potential. Likewise, highlight the type of diseases caused
by Staphylococcus aureus, which could be combated by the use of CMM extracts.

Answer: Done. The required table was added (table 7)

Comment 4: Additionally, it would be advisable to improve the wording of the conclusions. The results shown do not support the following conclusion: Besides, a considerable antioxidant activity and significant antibacterial activity of the methanolic extract against S. aureus were detected. The antioxidant activity evaluated only against the DPPH radical is not sufficient for such a conclusion. Answer: Done.

Round 2

Reviewer 2 Report

Comments and Suggestions for Authors

I consider that the paper was improved in its global presentation, so I recommend its publicaction in the current form.